# Evaluation of AIF-1 (Allograft Inflammatory Factor-1) as a Biomarker of Crohn’s Disease Severity

**DOI:** 10.3390/biomedicines10030727

**Published:** 2022-03-21

**Authors:** Luis G. Guijarro, David Cano-Martínez, M. Val Toledo-Lobo, Lidia Ruiz-Llorente, María Chaparro, Iván Guerra, Marisa Iborra, José Luis Cabriada, Luis Bujanda, Carlos Taxonera, Valle García-Sánchez, Ignacio Marín-Jiménez, Manuel Barreiro-de Acosta, Isabel Vera, María Dolores Martín-Arranz, Francisco Mesonero, Laura Sempere, Fernando Gomollón, Joaquín Hinojosa, Sofía Zoullas, Jorge Monserrat, Cesar Menor-Salvan, Melchor Alvarez-Mon, Javier P. Gisbert, Miguel A. Ortega, Borja Hernández-Breijo

**Affiliations:** 1Department of System Biology, University of Alcalá, 28805 Alcalá de Henares, Madrid, Spain; luis.gonzalez@uah.es (L.G.G.); david.cano@uah.es (D.C.-M.); lidia.ruizl@uah.es (L.R.-L.); sofiazoullas@gmail.com (S.Z.); cesar.menor@uah.es (C.M.-S.); 2Centro de Investigación Biomédica en Red de Enfermedades Hepáticas y Digestivas (CIBEREHD), 28029 Madrid, Madrid, Spain; jorge.monserrat@uah.es (J.M.); mademons@gmail.com (M.A.-M.); 3Ramón y Cajal Institute of Sanitary Research (IRYCIS), 28034 Madrid, Madrid, Spain; 4Department of Biomedicine and Biotechnology, University of Alcalá, 28805 Alcalá de Henares, Madrid, Spain; mval.toledo@uah.es; 5Gastroenterology Unit, Hospital Universitario de La Princesa, Instituto de Investigación Sanitaria Princesa (IIS-IP), Universidad Autónoma de Madrid, 28049 Madrid, Madrid, Spain; maria.chaparro@salud.madrid.org; 6Gastroenterology Unit, Hospital Universitario de Fuenlabrada, Instituto de Investigación Hospital Universitario La Paz (IdiPaz), 28029 Fuenlabrada, Madrid, Spain; ivangm79@gmail.com; 7Gastroenterology Unit, Hospital Universitario de La Fe (CIBEREHD), 46026 Valencia, Valencia, Spain; marisaiborra@hotmail.com; 8Gastroenterology Unit, Hospital Universitario de Galdakano, 48960 Galdakao, Vizcaya, Spain; jcabriada@gmail.com; 9Gastroenterology Unit, Hospital Universitario de Donostia, 20014 San Sebastián, Guipúzcoa, Spain; luis.bujandafernandezdepierola@osakidetza.eus; 10Gastroenterology Unit, Hospital Universitario Clínico San Carlos, IdISSC, 28040 Madrid, Madrid, Spain; carlos.taxonera@salud.madrid.org; 11Gastroenterology Unit, Hospital Universitario Reina Sofía, 14004 Córdoba, Córdoba, Spain; vallegarciasanchez@gmail.com; 12Gastroenterology Unit, Hospital Universitario Gregorio Marañón, IiSGM, 28007 Madrid, Madrid, Spain; drnachomarin@hotmail.com; 13Gastroenterology Unit, Hospital Universitario Clínico de Santiago, 15706 Santiago de Compostela, La Coruña, Spain; manubarreiro@hotmail.com; 14Gastroenterology Unit, Hospital Universitario Puerta de Hierro Majadahonda, 28222 Majadahonda, Madrid, Spain; isabel.veramendoza@gmail.com; 15Gastroenterology Unit, Hospital Universitario La Paz, 28046 Madrid, Madrid, Spain; mmartinarranz@salud.madrid.org; 16Gastroenterology Unit, Hospital Universitario Ramón y Cajal, 28034 Madrid, Madrid, Spain; pacomeso@hotmail.com; 17Gastroenterology Unit, Hospital Universitario Alicante, 03010 Alicante, Alicante, Spain; lausemro@hotmail.com; 18Gastroenterology Unit, Hospital Clínico Universitario Lozano Blesa, IIS Aragón, 50009 Zaragoza, Zaragoza, Spain; fgomollon@gmail.com; 19Gastroenterology Unit, Hospital Universitario Manises, 46940 Valencia, Valencia, Spain; jhinojosad@gmail.com; 20Department of Medicine and Medical Specialities, Faculty of Medicine and Health Sciences, University of Alcalá, 28801 Alcalá de Henares, Madrid, Spain; 21Immune System Diseases-Rheumatology, Oncology Service and Internal Medicine, University Hospital Príncipe de Asturias, 28805 Alcalá de Henares, Madrid, Spain; 22Immuno-Rheumatology Research Group, Instituto de Investigación Hospital Universitario La Paz (IdiPaz), 28046 Madrid, Madrid, Spain; borja.hernandez@idipaz.es

**Keywords:** AIF-1, Crohn’s disease, anti-TNFs, CRP

## Abstract

Background: Recently, increased tissue levels of AIF-1 have been shown in experimental colitis, supporting its role in intestinal inflammation. Therefore, we studied the levels of AIF-1 in Crohn’s disease (CD). Methods: This study included 33 patients with CD (14 men and 19 women) who participated in the PREDICROHN project, a prospective multicenter study of the Spanish Group of Inflammatory bowel disease (GETECCU). Results: This article demonstrates declines with respect to baseline levels of serum AIF-1 in Crohn’s disease (CD) patients after 14 weeks of treatment with anti-TNFs. Furthermore, in patients with active CD (HB ≥ 5), serum AIF-1 levels were significantly higher than those in patients without activity (HB ≤ 4). The study of serum AIF-1 in the same cohort, revealed an area under the ROC curve (AUC) value of AUC = 0.66 (*p* = 0.014), while for the CRP (C-reactive protein), (AUC) value of 0.69 (*p* = 0.0066), indicating a similar ability to classify CD patients by their severity. However, the combination of data on serum levels of AIF-1 and CRP improves the predictive ability of these analyses for classifying CD patients as active (HB ≥ 5) or inactive (HB ≤ 4). When we used the odds ratio (OR) formula, we observed that patients with CRP > 5 mg/L or AIF-1 > 200 pg/mL or both conditions were 13 times more likely to show HB ≥ 5 (active CD) than were those with both markers below these thresholds. Conclusion: The development of an algorithm that includes serum levels of AIF-1 and CRP could be useful for assessing Crohn’s disease severity.

## 1. Introduction

The incidence and prevalence of Crohn’s disease (CD) have been increasing annually worldwide [1]. In CD, inflammation affects all intestinal layers [2] and leads to increases in several inflammatory cytokines, such as TNF-α [3]. Blocking TNF-α using specific antibodies, such as infliximab (IFX) or adalimumab (ADA), has demonstrated its effectiveness in several clinical trials [4,5]. However, the clinical response to TNF-α blockers is quite variable. Laboratory and genetic markers have been proposed as indicators of response to biologics. For instance, high serum C-reactive protein (CRP) has been associated with a favorable response to IFX, whereas a combination of serological markers, positive p-anti-neutrophil cytoplasmic antibody (ANCA) and negative anti-*Saccharomyces cerevisiae* antibody (ASCA), have been associated with poor response [6]. Therefore, the search for new biomarkers that indicate the evolution of CD is a priority objective. AIF-1 (allograft inflammatory factor-1) is a 17 kDa cytosolic protein produced by monocytes, macrophages, and lymphocytes that can be secreted to the outside of the cell and can act as a cytokine [7]. It was initially described for its ability to cause cardiac allograft rejection [8]. Subsequently, its involvement in rheumatoid arthritis and atherosclerosis has been revealed [9,10]. Recent studies have shown that AIF-1 acts as a chemoattractant for macrophages [11] as well as a promotor of T-cell proliferation [12] and differentiation toward the Th1 phenotype [13]. Likewise, its inhibitory role in the Treg response has been demonstrated [14,15]. All these data seem to indicate that this protein plays a proinflammatory role, acting on the cells of the immune system and through those in inflamed tissues. Furthermore, we recently demonstrated an increase in tissue levels of AIF-1 in rats with experimental colitis induced with DSS (dextran sodium sulfate), supporting its role in intestinal inflammation [16]. Since DSS-induced colitis is a good model to study CD in humans, we decided to evaluate the capacity of AIF-1 alone or in combination with CRP (a classical biomarker of the disease) to predict the severity of CD. This objective was pursued by the analysis of serum AIF-1 levels in CD patients treated with TNF-α blockers. Likewise, the levels of AIF-1 were studied in colon biopsies of patients with CD.

## 2. Materials and Methods

### 2.1. Patients

In this study, 33 patients with CD (14 men and 19 women) who participated in the PREDICROHN project, a prospective multicenter study of the Spanish Inflammatory bowel disease (IBD) Group (GETECCU), were included. The patients who were assessed fulfilled the following criteria: over 18 years of age; diagnosed with luminal or perianal CD using clinical, radiological, endoscopic, and histological methods; and required anti-TNF treatment. The exclusion criteria were as follows: pregnant or nursing women; patients infected with HBV, HCV, or HIV; and patients under previous treatment with anti-TNF or other biological drugs. All patients naïve to anti-TNF therapy were assigned to receive adalimumab (ADA) or infliximab (IFX) according to standard clinical practice and were followed-up for 1 year. Patients received 160/80 mg adalimumab at weeks 0 and 2 and 40 mg at week 6 and every 8 weeks thereafter (14, 22, 30, 38, 46, 54 weeks) or infliximab 5 mg/kg at weeks 0, 2, 6, and every 8 weeks thereafter (14, 22, 30, 38, 46, 54 weeks). Patient sera were analyzed at baseline (week 0) and at 14 weeks after the beginning of anti-TNF administration. Disease activity was evaluated by the Harvey-Bradshaw (HB) index. An HB index ≤ 4 was considered a clinical remission.

In a restricted number of patients, a biopsy was taken during endoscopy of inflamed colon tissue (Crohn’s disease-affected region) and healthy adjacent tissue.

The present study was performed in accordance with the guidelines of the Helsinki Declaration of 1975. The project obtained approval of Institutional Ethics Committees from all hospitals involved. All patients signed informed consent at the time of their clinical evaluations.

### 2.2. Experimental Colitis Induced by DSS

Male Wistar rats weighing 180 to 200 g and aged 12 weeks were housed in the Animal House at the University of Alcalá under temperature- and light-controlled conditions (23 ± 1.5 °C). Animals were handled according to the criteria described in the “Guide for the Care and Use of Laboratory Animals” edited by the National Academy of Sciences and published by the National Institutes of Health (Office of Laboratory Animal Welfare) https://olaw.nih.gov/resources/publications/guide-care-2011.htm, accessed on 15 January 2022. The present study was performed in accordance with protocols approved by the Ethics Committee on Animal Experiments at Alcala University. The animals were fed a standard diet (2.9 kcal/g, 4% fat, 14.3% protein) throughout their lifespans. The animals were separated into two groups: a healthy group and an experimental colitis group. Experimental colitis was induced by treating animals with 5% dextran sulfate sodium (DSS) dissolved in drinking water [17] for five days. In each experimental group, 5–6 animals were used. All animals were sacrificed on the fifth day of experimental conditions.

### 2.3. General Assessment of Colitis

To characterize the extent of the disease, we determined daily body weight evolution and the presence or absence of diarrhea and blood in stools. After sacrifice, we determined the biochemical parameters that are summarized in Table 1, and we performed histological determinations.

### 2.4. Biochemical Parameters

Blood samples for biochemical analysis were collected from the rats at the time of sacrifice by cardiac puncture. Serum was collected following 30 min incubation at room temperature and 5 min of centrifugation at 500× *g*. Biochemical parameters were analyzed using a hematologic analyzer. Determination of serum IFX levels was performed by ELISA as previously described [16].

### 2.5. Tissue Preparation, Immunohistochemistry, and Immunofluorescence Staining

The preparation of the tissue for the immunohistochemical study was carried out by fixing colon samples with neutral formalin, washing with PBS, dehydrating samples with a graded ethanol series, and embedding the samples in paraffin. Five-micron thick sections were obtained and mounted on silanized glass slides. The periodic acid-Schiff (PAS) reaction and Alcian blue staining were used to evaluate colon mucosal integrity, to visualize goblet cells, and, above all, to differentiate neutral and acidic mucin as previously described [16].

For immunohistochemical studies, we used rabbit anti-β-catenin (Santa Cruz Biotechnology, Santa Cruz, CA, USA) or rabbit anti-AIF-1 (Wako, Osaka, Japan). After incubation with the primary antibody, the samples were incubated with peroxidase-conjugated secondary antibody (polymer-based MasVision^TM^ system from Master Diagnostica, Granada, Spain). Nuclei were stained with Carazzi’s hematoxylin. Negative controls were performed by incubating the samples with nonimmune mouse serum instead of the primary antibody.

To analyze apoptotic nuclei, the DeadEndFluorometric TUNEL System (Promega, Madrid, Spain) was used according to the manufacturer’s instructions. Briefly, deparaffinized and hydrated sections were washed with a 0.85% solution of NaCl at room temperature for five minutes and sequentially washed with PBS (5 min), 4% methanol (15 min), and PBS again (2 washes, 5 min each). After protein digestion (proteinase K, 20 µg/mL) and a new round of PBS and methanol washes, samples were treated with equilibration buffer for 7 min and incubated with the buffered nucleotide mixture and the TdT enzyme (at 37 °C for 60 min), with all materials supplied in the kit. Then, sections were washed with double-distilled water and cover-slipped using FluoroGuard^TM^ (Bellevue, WA, USA) non-fade mounting media with Hoechst 33342 (Bio-Rad, Hercules, CA, USA). Slides were examined on an epifluorescence microscope (Olympus BX50, Tokyo, Japan).

### 2.6. Protein Extraction and Immunoblotting

Rat colonic tissue was homogenized in ice-cold lysis buffer (50 mM Tris-HCl, pH 7.4, 5 mM EDTA, 1 mM EGTA, 1 mM PMSF, 5 μg/mL leupeptin, and 5 μg/mL aprotinin) by mechanical disruption. To remove connective tissue, centrifugation (500 rpm for 10 min at 4 °C) was carried out. Each supernatant was collected and stored at −80 °C until use. The amount of protein present was determined using a Bradford protein assay kit (Bio-Rad). Protein extracts were analyzed by SDS–PAGE and immunoblots were carried out as previously described [16] using primary antibodies against AIF-1 (Wako, Osaka, Japan), AKT (Cell Signaling Technology, Danvers, MA, USA), and albumin (Sigma-Aldrich, St. Louis, MO, USA) for serum specimens.

### 2.7. Isolation of Mononuclear Cells from Human Peripheral Blood

Blood samples were collected by antecubital puncture from six healthy volunteers (5 males and 1 female) in the consulting room of Principe de Asturias University Hospital. A consent form was obtained from the participants in the study. PBMCs were purified from heparinized venous blood by Ficoll-Hypaque density gradient centrifugation. Cells were resuspended in RPMI-1640 medium supplemented with 10% heat-inactivated fetal bovine serum (FBS), 25 mM HEPES and 1% penicillin-streptomycin.

### 2.8. Real Time Quantitative PCR Protocol

PBMCs were stimulated with phorbol 12-myristate 13-acetate (PMA) as previously described [13]. Then, total RNA was extracted from the PBMCs using an RNeasy Mini Kit following the manufacturer’s protocol. RNA quality and concentration were estimated by optical density measurement using a Nanodrop. Total RNA (500 ng) underwent reverse transcription (RT) to complementary deoxyribonucleic acid (cDNA). RT was performed at 37 °C for 15 min and at 85 °C for 5 s using the PrimeScript^®^ RT reagent Kit (Takara Bio, Saint-Germain-en-Laye, France) following the manufacturer’s protocol. Quantitative PCR was carried out using a Kapa Sybr Fast qPCR Kit (Merck, Darmstadt, Germany) and the following primers:

IL-2 (forward) 5′ CACTAATTCTTGCACTTGTCAC 3′IL-2 (reverse) 5′ CCTTCTTGGGCATGTAAAACT 3′IFN-gamma (forward) 5′ CTAATTATTCGGTAACTGACTTGA 3′IFN-gamma (reverse) 5′ ACAGTTCAGCCATCACTTGGA 3′AIF-1 isoform 2 (forward) 5′ ATGGAGTTTGACCTTAATGGAAATGGC 3′AIF-1 isoform 2 (reverse) 5′ TCACATTTTTAGGATGGCAGACCTCTTG 3′18-S (forward) 5′ GGACACGGACAGGATTGACA 3′18-S (reverse) 5′ ACCCACGGAATCGAGAAAGA 3′

### 2.9. Statistical Analysis

Descriptive analyses were performed for the variables studied. The results are presented as the mean ± SEM or mean ± SD (or median and interquartile range [IQR]) for continuous variables and absolute numbers and relative frequencies for categorical variables. Comparisons of unpaired continuous data were conducted using the unpaired *t*-test or Mann–Whitney *U* test, depending on how the data were distributed. Comparisons of paired continuous data were conducted using the paired *t*-test or Mann–Whitney *U* test, depending on the data distribution. For multiple comparisons, one-way ANOVA plus a post hoc Tukey’s multiple test was performed. Correlations of AIF levels with HB or CRP levels were analyzed using Pearson’s correlation coefficient (R). Receiver operating characteristic (ROC) analyses were performed to determine associations of AIF-1 and CRP levels with clinical remission by HB. A *p*-value < 0.05 was considered statistically significant (* *p* < 0.05; ** *p* < 0.01; *** *p* < 0.001).

## 3. Results

### 3.1. Changes Observed in AIF-1 Levels during DSS-Induced Colitis in Rats

One group of rats was treated with DSS in drinking water for 5 days (DSS group), while the other group received tap water ad libitum (control group). During this period of time, the animals were weighed at the beginning of each day. The results obtained are shown in Figure 1A. The control group gained body weight for the five days of assessment, while the group treated with DSS during this period did not. On the fifth day of treatment, the animals were sacrificed, and serum samples and the central part of the colon were obtained for histological studies. We observed a decrease in circulating glucose levels, as well as an increase in TNF-α levels, in the DSS group compared to the controls (Figure 1B and Table 1). However, we did not observe significant changes in the levels of albumin, total proteins, or aminotransferases (AST, ALT), which seems to indicate that there was no significant liver involvement during this period of time. Direct observation of the animals in the DSS group showed strong reddening of the anus, together with diarrhea and blood in the stool, which suggests a diagnosis of hemorrhagic colitis.

The histological analyses confirmed a diagnosis of inflammation of the colon with a decrease in mucin-producing cells (PAS-Alcian blue staining) and a weakening of the intercellular junctions (beta-catenin staining) in the apical zone of the Lieberkühn glands, together with an increase in apoptosis (TUNEL staining) of the cells present in the mucosa and submucosa (Figure 1C) in rats in the DSS group.

All these data suggest a weakening of the intestinal barrier, which explains the presence of inflammatory colitis. To confirm this hypothesis, we studied the levels of AIF-1 in the colon and in the serum of animals treated with DSS and compared them with those of control animals (Figure 2). Histological studies revealed that in control animals, AIF-1 was found to a greater extent in the region of the mucosa just below the epithelial cells, forming a barrier (Figure 2A, inset). Occasionally, we found cells that contained AIF-1 in the muscle area, mainly in the circular muscle (CM) region characterized by cells with elongated nuclei and to a lesser extent in the longitudinal muscle (LM) in which rounded nuclei are observed (Figure 2A).

However, in the rats belonging to the DSS group, we found numerous ulcers characterized by the infiltration of immune cells containing AIF-1 (Figure 2A). When analyzing this inflamed area with higher magnification, we observed that the muscle layer was highly infiltrated with AIF-1-positive cells. As we can see in Figure 2A, the amplified region presents muscle cells with elongated nuclei, which suggests a circular muscle region. (Figure 2A). Increased levels of AIF-1 in experimental colitis were confirmed by immunoblotting of AIF-1 in colon homogenates from DSS-treated animals (Figure 2B) and in serum samples (Figure 2C). As loading controls, the levels of ERK1/2 and albumin were assessed in the tissue and serum samples, respectively.

### 3.2. Changes Observed in AIF-1 Expression in the Colon of Patients with Crohn’s Disease and in the Levels of AIF-1 in PBMCs Activated by PMA

AIF-1 was higher in the ulcerated colon epithelium of Crohn’s disease patients than in corresponding healthy epithelium from the same patients (Figure 3A). Figure 3A presents the results from two Crohn’s patients, which are representative of five who were studied. To demonstrate the immunological origin of AIF-1, we examined the changes in the protein levels as a result of nonspecific activation of leukocytes by PMA (a PKC activator). For this, peripheral blood leukocytes (PBMCs) were isolated from control individuals and activated with PMA (50 ng/mL) for 24 h at 37 °C, in a humidified 5% CO_2_ environment. As shown in Figure 3B, the activation of the PKC enzymes produced strong increases in transcriptional expression of IL-2, IFN-gamma, and AIF-1. The latter protein was also assessed in the extracellular medium and in the cell lysates from PBMCs by immunoblotting (Figure 3C). As a positive control, we incubated PBMCs in the same conditions with CD3 CD28 Dynabeads. Therefore, AIF-1 was able to be observed in inflamed tissues, in serum, and in cells of the immune system, and we used this capability to study its evolution in serum in patients with Crohn’s disease treated with anti-TNF.

### 3.3. Evaluation of AIF-1 as Biomarker of Crohn’s Disease Severity

As a preliminary step, we studied the evolution of the disease activity index as measured by the Harvey-Bradshaw (HB) index scores of these patients treated with infliximab or adalimumab for 54 weeks. Treatment with these drugs produced a significant reduction in the HB index from the second week, which was maintained throughout the treatment period (Figure 4). The assessment of the levels of AIF-1 and its correlation with other biochemical parameters was carried out with the samples of the patients at baseline and after 14 weeks of treatment. At this period of time, there was a significant decrease in AIF-1 with respect to the baseline situation (Figure 5A), as occurs with a classic biomarker such as CRP (Figure 5A). When the patients were classified according to the HB scores following previously established criteria into inactive or quiescent (HB ≤ 4) and active patients (HB ≥ 5), we observed that the latter group presented higher levels of AIF-1 and CRP than levels in patients with inactive disease (Figure 5B). Likewise, a positive and statistically significant correlation was observed between the HB and AIF-1 values (Figure 5C). For the ROC curves of sensitivity and diagnostic specificity of AIF-1 for Crohn’s disease severity as assessed by the HB index, we observed that the AUC was 0.66 (*p* = 0.014) (Figure 5D), which was very similar to that obtained with CRP (AUC = 0.69, *p* = 0.0066) (Figure 5D). Since neither biomarker exhibited very high diagnostic sensitivity or specificity values separately, we wanted to see if performance was improved when they were combined. We first studied the existing correlation between the two parameters, and we found a statistically significant positive linear correlation between the two analytes (Pearson’s coefficient = 0.385, *p* < 0.007) (Figure 6). In this graph, active patients (HB ≥ 5) are shown in red and inactive patients (HB ≤ 4) are shown in blue. To determine the predictive capacity of AIF-1 and CRP together, we used the odds ratio (OR) methodology. We consider that patients at high risk of suffering from active CD (HB ≥ 5) were those who showed levels of CRP > 5 mg/L or AIF-1 > 200 pg/mL or both conditions at the same time. Those who were below the thresholds for both analytes had low risk of presenting active Crohn’s disease. We chose the CRP threshold following the international criteria in a previously published meta-analysis [18]. Given that for AIF-1, the studies necessary to establish a threshold have not yet been carried out, we consider for this purpose the mean ± 3 SD of non-active patients (HB ≤ 4), which represents the 99.7% confidence limit (Figure 5B). Under these criteria, we obtained an OR = 13 (*p* = 0.0014), which indicates that patients with elevated AIF-1, CRP, or both analytes were nearly 13 times more likely to have active CD (HB ≥ 5) than those with both markers below these thresholds. Interestingly, when we calculated the OR for AIF-1 (cut-off 200 pg/mL) and CRP (cut-off 5 mg/L) separately, the results were worse than for the combined biomarkers. We have obtained the following results for AIF-1 (OR = 7.74; 95% CI (0.88–67.69); *p* = 0.064 and for CRP (OR = 7.57; 95% CI (1.9–30.18; *p* = 0.0041). Table 2 presents the statistical data of the patients studied. After 14 weeks of treatment with anti-TNFs, there were significant declines in numbers of leukocytes, neutrophils, and platelets, as well as in the inflammatory indices (CRP, AIF-1, and ESR). A significant decrease in fibrinogen levels was also observed.

## 4. Discussion

In this study, we have demonstrated the presence of AIF-1 in the rat and human colon mucosa and that its expression increases in this intestinal region during experimental colitis or in Crohn’s disease, respectively. AIF-1 has previously been found in T lymphocytes [19] and in other human peripheral blood cells [7]. In both species, it is located just below the epithelial cells of the colon and distributed almost continuously along the epithelial barrier, suggesting its protective role. In this sense, it has been observed that mice that overexpress AIF-1 are more resistant to colitis induced by TNBS [20].

The expression of AIF-1 could be stimulated in circulating human blood leukocytes (PBMCs) by incubation of these cells with PKC activators (such as PMA). Previously, we have shown that PBMCs can release AIF-1 under CD3-CD28 stimulation [13]. In addition, exogenous AIF-1 can increase INF-γ levels in PBMCs [13]. These data suggest that the release of AIF-1 to circulation could play a proinflammatory role and therefore could be involved in the pathological mechanism underlying inflammatory bowel diseases. In Crohn’s patients, we observed an increase in AIF-1 in the damaged area of the colon with respect to levels in the surrounding normal area. The functional significance of the increase in AIF-1 in inflamed tissue is currently unknown; some authors suggest a protective role has already been indicated [20]. However, others argue that AIF-1could be a factor that enhances inflammation since it favors the differentiation of the Th1 type [13] and activates the invasive capacity of lymphocytes [19] and monocytes [11]. To analyze the possible usefulness of AIF-1 as a biomarker in Crohn’s disease, we studied the levels of AIF-1 in a cohort of naïve patients treated with anti-TNF (infliximab or adalimumab). We observed that these drugs significantly decreased the Harvey-Bradshaw (HB) index scores after 2 weeks of treatment and that the effect was maintained for at least 54 weeks. We subsequently studied serum levels of AIF-1 at baseline and at 14 weeks, as well as other biomarkers of proven clinical relevance. Over this period of time, improvement of patients was accompanied by significant declines in the levels of AIF-1 and CRP, as previously observed for the latter protein [21]. Likewise, our study shows that there is a positive and statistically significant correlation between HB and serum level of AIF-1 in Crohn’s disease patients. Previous studies have found strong correlations of levels of fecal lactoferrin and calprotectin with the CDAI index [22]. When we stratified the patient cohort by HB scores, we observed that active patients (HB ≥ 5) showed higher serum levels of AIF-1 and CRP than did inactive patients (HB ≤ 4), which indicates their potential usefulness in monitoring CD.

The ROC curves constructed for AIF-1 and CRP were very similar and demonstrate a discrete diagnostic sensitivity and specificity as previously described for CRP in CD [23] so we think that the study of both analytes could have a value added in the monitoring of this pathology.

The two analytes, AIF-1 and CRP, seem to be significantly correlated in serum, as we were able to verify by examining Pearson’s correlation coefficient (R = 0.385, *p* < 0.007). When we separated the patients into inactive (HB ≤ 4, blue) and active (HB ≥ 5, red) groups, we observed that most of the active patients exhibited either elevated CRP (>5 mg/L), elevated AIF-1 (>200 pg/mL) or both, the patients who we called the high-risk group. Those with values below the thresholds for both parameters were called the low-risk group. Calculating the OR between both groups, we observed that the high-risk group had a 13.36 higher probability of having active Crohn’s disease than did the low-risk group. As indicated previously, we chose the CRP threshold based on previous data in the literature [18]. For AIF-1, we used as criteria the 99.7% confidence limits of inactive patients, which represent mathematically the value of the mean ± 3 SD of non-active patients. The predictive ability for CD severity using the biomarkers AIF-1 or CRP separately was minor with respect to the method in which both were considered together. The mechanistic role played by serum AIF-1 in Crohn’s disease currently remains unknown. Although, as previously indicated [11,12,13,14,15,19], numerous effects of AIF-1 have been observed on the cells of the immune system, as of now, we do not know if it favors the spread of inflammation in CD patients, which would make AIF-1 a possible therapeutic target. In any case, the assessment of AIF-1 in serum could be of interest to monitor Crohn’s disease in patients, especially in whom CRP does not increase due to the existence of low-expression SNP polymorphisms [24]. The polymorphisms rs1205 and rs3093059 were significantly associated with CRP levels [24]. In inflammatory bowel disease patients, three single nucleotide polymorphisms (rs1205, rs1130864, and rs1417938) showed association with elevated CRP levels at diagnosis [25]. In these cases, patients could show only moderate increases in CRP despite exhibiting high inflammatory activity. We are aware that the present work is a pilot study and that with a view to proposing AIF-1 as a new CD biomarker, it would be essential to carry out new studies with a broader population of CD patients.

## 5. Conclusions

In conclusion, our results suggest that AIF-1 could be a complementary biomarker of CRP to monitor the severity of Crohn’s disease.

## Figures and Tables

**Figure 1 biomedicines-10-00727-f001:**
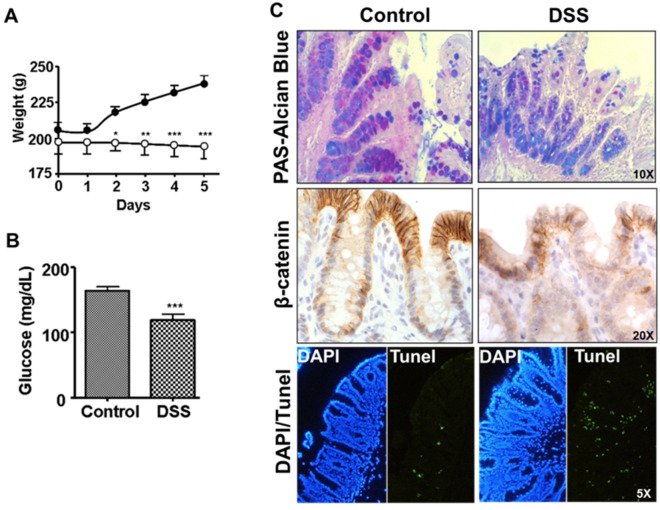
General analysis of the state of the rats with colitis. (**A**) Evolution of the weight of the animals during the colitis induced by DSS. (**B**) Blood glucose levels after 5 days of treatment with DSS. (**C**) Histological study of the colon after 5 days of treatment with DSS. The expression of mucin, β-catenin levels and apoptosis were studied. Significance levels: * *p* < 0.05; ** *p* < 0.01; *** *p* < 0.001.

**Figure 2 biomedicines-10-00727-f002:**
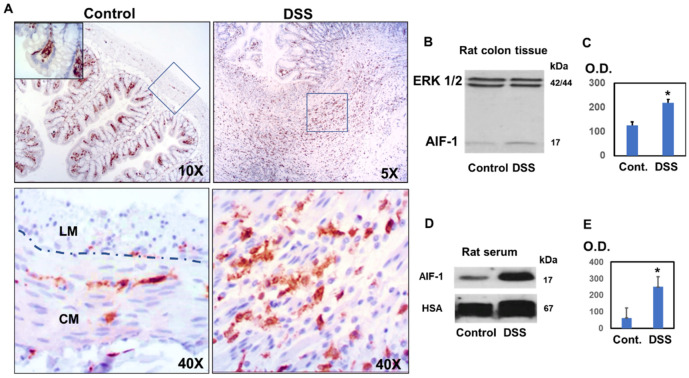
Study of the levels of AIF-1 in the colon and in the serum of rats with colitis induced by DSS for 5 days. (**A**) Localization of AIF-1 by immunohistochemical techniques in the colon of rats treated with DSS. The region indicated by the square has been enlarged in the figure below. (**B**) Analysis of AIF-1 levels in the colon by immunoblot. ERK has been studied as a load control. (**C**) Densitometric analysis of AIF-1 levels in the colon obtained by immunoblot. Mean ± SEM. (**D**) Study of AIF-1 levels in serum by immunoblot. Human serum albumin (HSA) has been tested as a loading control. (**E**) Densitometric analysis of AIF-1 levels in the serum obtained by immunoblot. Mean ± SEM. The images are representative of at least 5 others with similar results. LM = longitudinal muscle region. CM = circular muscle region. O.D. = optical density. Significance levels: * *p* < 0.05.

**Figure 3 biomedicines-10-00727-f003:**
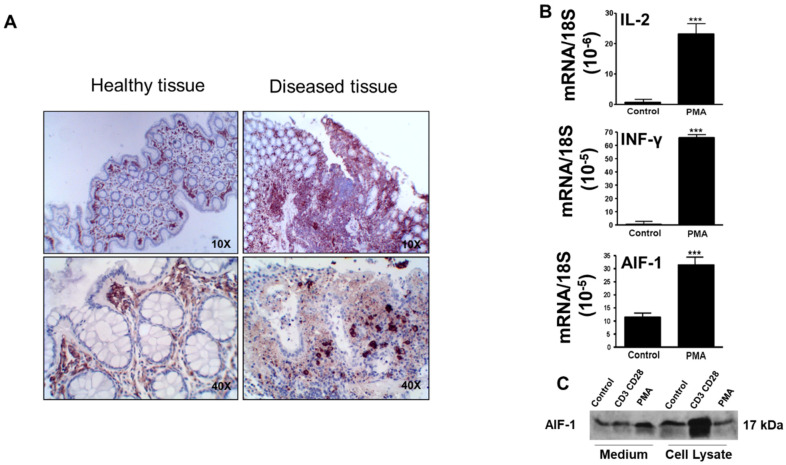
AIF-1 levels in human colon biopsies from CD patients and in human PBMC after stimulation with PMA. (**A**) Comparison of AIF-1 levels between the affected and healthy areas in colon biopsies of patients with CD. (**B**) Study of AIF-1 mRNA levels in PBMCs after stimulation with PMA. The results have been compared with the other cytokines (IL-2 and INF-γ). (**C**) AIF-1 protein levels in PBMCs after the stimulus with PMA. Significance levels: *** *p* < 0.001.

**Figure 4 biomedicines-10-00727-f004:**
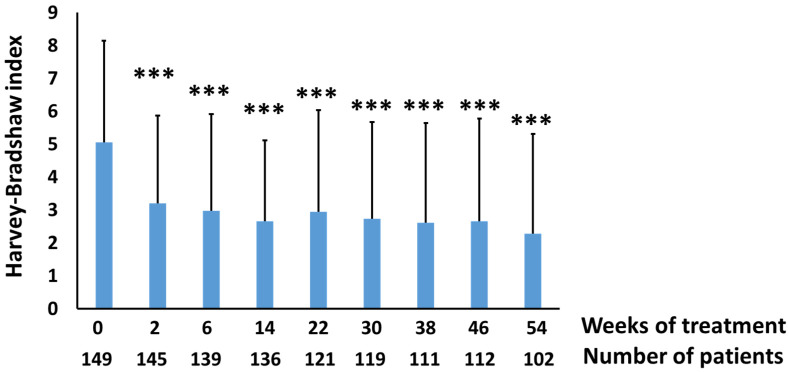
Harvey-Bradshaw index evolution of patients with Crohn’s disease during anti-TNF treatment. Significance levels: *** *p* < 0.001.

**Figure 5 biomedicines-10-00727-f005:**
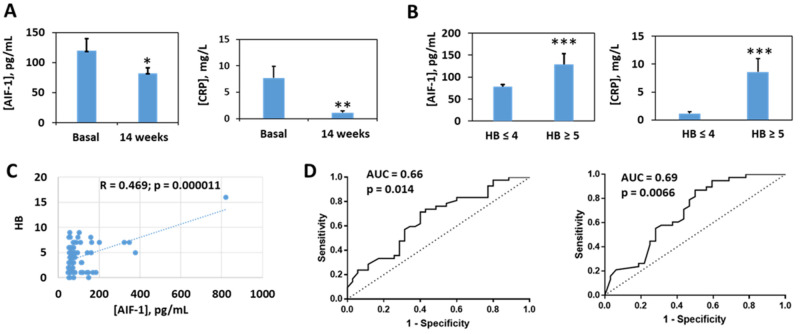
Comparison of serum AIF-1 and CRP levels in Crohn’s disease patients. (**A**) Levels of both markers in serum at baseline and at 14 weeks of anti-TNFα treatment. The mean ± SEM is shown. (**B**) Comparison of the levels of both serum markers between inactive (HB ≤ 4) and active (HB ≥ 5) patients. The mean ± SEM is shown. (**C**) Correlation between serum AIF-1 levels and the HB index of patients with Crohn’s disease. (**D**) ROC curves for AIF-1 (**left**) and CRP (**right**). Significance levels: * *p* < 0.05; ** *p* < 0.01; *** *p* < 0.001.

**Figure 6 biomedicines-10-00727-f006:**
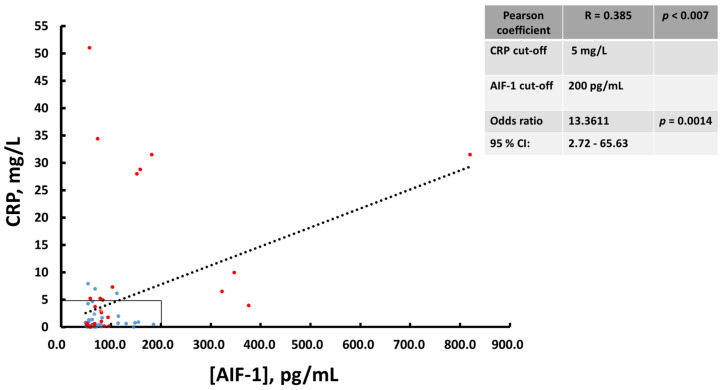
Correlation between serum levels of AIF-1 and CRP in patients with Crohn’s disease. The red dots represent patients with HB ≥ 5 and the blue dots those with HB ≤ 4. The regions that were compared to obtain the OR were high risk (CRP > 5 mg/L or AIF-1 > 200 pg/mL or both conditions at the same time) and low risk (CRP ≤ 5 mg/L and AIF-1 ≤ 200 pg/mL) for active (HB ≥ 5) or inactive (HB ≤ 4) CD.

**Table 1 biomedicines-10-00727-t001:** Biochemical parameters in serum obtained from rat with colitis induced by DSS for 5 days. Healthy rats (Control group) and rats subjected to experimental colitis (treated with DSS). Values are the mean ± SD. Data were analyzed by student’s *t*-test. (* *p* < 0.05, ** *p* < 0.01).

	Control	DSS
Glucose (mg/dL)	169.7 ± 11.4	112.5 ± 12.9 *
Albumin (g/dL)	1.30 ± 0.12	1.20 ± 0.10
Total proteins (g/dL)	5.63 ± 0.13	5.57 ± 0.20
AST (U/L)	85.0 ± 10.6	87.5 ± 5.95
ALT (U/L)	28.8 ± 6.40	29.5 ± 7.86
TNF-α (pg/mL)	131.0 ± 25.0	340.1 ± 30.0 **
Diarrhoea	No	Yes
Bleeding	No	Yes

**Table 2 biomedicines-10-00727-t002:** Hematological indices and biochemical parameters obtained from patients with CD under baseline conditions and after 14 weeks of anti-TNFα treatment. * *p* < 0.05; ** *p* < 0.01; *** *p* < 0.001.

	Basal	14 Weeks	*p*-Value
N = 33			
Age (years)	44.1 (20–67)		
Male, n (%)	14 (42.4)		
Smokers, n (%)	11 (33.3)		
Weight (kg)	68.9 (45–98)	67.8 (45–97)	
Height (cm)	166.9 (150–188)	166.9 (150–188)	
Harvey-Bradshaw	5.4 ± 3.1	2.9 ± 1.8 *	0.019
Haematological indices			
Leukocytes (10^9^/L)	7.2 ± 3.4	5.9 ± 2.3 **	0.0011
Neutrophils (10^9^/L)	5.3 ± 3.3	3.6 ± 1.8 **	0.0011
Platelets (10^9^/L)	321.9 ± 100.8	278.9 ± 63.8 **	0.001
Haemoglobin (g/dL)	13.2 ± 1.3	13.4 ± 1.3	0.126
Ferritin (ng/mL)	67.3 ± 63.5	51.9 ± 69.7	0.123
Inflammatory indices			
CRP (mg/L)	8.2 ± 13.3	1.2 ± 1.9 **	0.0023
ESR (mm/h)	26.6 ± 19.4	11.9 ± 11.4 ***	6.8 × 10^−5^
AIF-1 (pg/mL)	119.4 ± 129.5	81.9 ± 57.2 *	0.024
Hepatic biomarkers			
Albumin (mg/dL)	4.15 ± 0.36	4.27 ± 0.36 *	0.0143
Fibrinogen (mg/dL)	460.4 ± 113.9	333.7 ± 88.2 ***	8.7 × 10^−5^

## Data Availability

Data from this study are available at the University of Alcalá and will be made available upon request.

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
