# Peer review of "Evaluation of AIF-1 (Allograft Inflammatory Factor-1) as a Biomarker of Crohn’s Disease Severity"

_biomedicines, 2022, doi:10.3390/biomedicines10030727_

Round 1

Reviewer 1 Report

very well written and documented manuscript. No further comments

Author Response

R1.- Very well written and documented manuscript. No further comments

 Thank you very much for your warm comment.

Reviewer 2 Report

General Comments:

In this manuscript, the authors clearly demonstrated that allograft inflammatory factor-1 (AIF-1) increased in the colon tissue of both rats and humans with active inflammation, and serum level of AIF-1 combined with serum CRP level was useful for monitoring the severity of Crohn’s disease (CD).

This is an excellent work that has provided the useful data for management of the patients with CD. However, several points as indicated below need to be addressed by authors to improve the quality of the article.

Major comments:

1) In the Figure 2A, the structures of the colon tissue were unclear. To describe the AIF-positive cells in the muscularis propria, the section stained with Hematoxylin and Eosin should be also shown.

2) If possible, please add the quantified data of Figure 2B.

3) The section “3.2 Comparison of AIF expression levels in rat and human colons.” and Figure 3. was not needed. The normal histology of the colon was shown in the Figure 2A Control and Figure 4A Healthy tissue. The pictures were only lined up and nothing was compared significantly.

4) In the section 3.4 Evaluation of AIF-1 ad biomarker of Crohn’s disease severity, the thresholds of CRP and AIF-1 might be used when AUC was calculated. The establishment of each cut-off value at the 13th to 15th sentences should be described before the statement about AUC.

5) The Odds ratio is most significant in this study in the Figure 7. Please add the conditions (“CRP > 5 mg/L or AIF-1 > 200 pg/mL or both conditions at the same time” in the text) also in the figure legend. And if possible, each Odds ratio of CRP or AIF-1 can be described respectively.

Minor comments:

1) The threshold of serum CRP and AIF-1 level should be consistent in the abstract (“≥ 5 mg/L, ≥ 200 pg/mL”) and the text (“> 5 mg/L, > 200 pg/mL”).

2) The Roman number “Table I.” at the second sentence in the section 2.3 may be alter to Arabic figure “Table 1.”

All points should be included in the manuscript.

Author Response

R2.- Comments and Suggestions for Authors

We effusively appreciate the constructive criticism of all reviewers that has allowed us to improve the article and we have taken it into account in its entirety. Below we have reviewed the comments made point by point and the subsequent response appears in the order shown

General Comments:

In this manuscript, the authors clearly demonstrated that allograft inflammatory factor-1 (AIF-1) increased in the colon tissue of both rats and humans with active inflammation, and serum level of AIF-1 combined with serum CRP level was useful for monitoring the severity of Crohn’s disease (CD).

This is an excellent work that has provided the useful data for management of the patients with CD. However, several points as indicated below need to be addressed by authors to improve the quality of the article.

Again, thank you very much for this general appreciation of the work.

Major comments:

1) In the Figure 2A, the structures of the colon tissue were unclear. To describe the AIF-positive cells in the muscularis propria, the section stained with Hematoxylin and Eosin should be also shown.

We have done a double AIF-1 & hematoxylin staining to better observe the presence of AIF-1, however following R2 instructions, we have replaced Figure 2A with a new one. The muscle area in the control and the inflamed region in the DSS group have been enlarged to better observe the presence of AIF-1-labeled cells. In control rats, the longitudinal muscle region (LM) is distinguished from the circular muscle region (CM) with a dashed line. All of these changes are commented on in the text.

2) If possible, please add the quantified data of Figure 2B.

We have carried out the densitometry of the western blots using the Scion Image program and two new histograms have been included, with the corresponding statistical study (Figures C and E).

3) The section “3.2 Comparison of AIF expression levels in rat and human colons.” and Figure 3. was not needed. The normal histology of the colon was shown in the Figure 2A Control and Figure 4A Healthy tissue. The pictures were only lined up and nothing was compared significantly.

We have removed this figure with the corresponding effect on the text.

4) In the section 3.4 Evaluation of AIF-1 ad biomarker of Crohn’s disease severity, the thresholds of CRP and AIF-1 might be used when AUC was calculated. The establishment of each cut-off value at the 13th to 15th sentences should be described before the statement about AUC.

We are sorry for not having responded adequately to this point but we did not fully understand the meaning of this phrase.

5) The Odds ratio is most significant in this study in the Figure 7. Please add the conditions (“CRP > 5 mg/L or AIF-1 > 200 pg/mL or both conditions at the same time” in the text) also in the figure legend. And if possible, each Odds ratio of CRP or AIF-1 can be described respectively.

We have added this phrase in the legend of figure 7 (now 6).

We have also calculated the ORs for AIF-1 and CRP separately, which have been included in the results and a comment in the discussion.

The values were:

OR (AIF-1): 7.74; CI (0.88-67.69); p = 0.064.

OR (CRP): 7.57; CI (1.90-30.18); p = 0.0041.

Minor comments:

1) The threshold of serum CRP and AIF-1 level should be consistent in the abstract (“≥ 5 mg/L, ≥ 200 pg/mL”) and the text (“> 5 mg/L, > 200 pg/mL”).

We have revised the manuscript throughout the text to take this comment into account.

2) The Roman number “Table I.” at the second sentence in the section 2.3 may be alter to Arabic figure “Table 1.”

We have made the change.

Finally, we want to say that we greatly appreciate the very constructive feedback we have received from reviewers.

Reviewer 3 Report

Although the study is trying to address an important and interesting subject, i.e. a new biomarker for patients with Crohn's disease, I did have difficulty in understanding the design and the rationale of the study while I was reading through it. Presicely

1) By reading the itroduction the authors do not mention anything about the experimental part of the animal study. On the contrary they mention "in this study we evaluated the capacity of AIF-1 and CRP to predict the severity of CD. This objective was pursued by the analysis of serum AFI-1 levels in CD patients treated with anti-TNFa blockers. Likewise the levels of AIF-1 were studied in colon biopsies of patients with CD". I do not see anything here about the animal models. I do not see how the aim of the study was structured and how the authors were planning to prove the relationship of CD with this novel biomarkers.

2) In the Methods section please provide information about the CD patients type of therapy and in whom were biopsies taken (how many patients since there were not taken in everyone) and with criteria were those patients selected

3) I find the Discussion confusing since data from results are presented here. The Discussion has to focus on what data are available about this subject so far and what new this study is bringing to the literature.

Author Response

R3.- Although the study is trying to address an important and interesting subject, i.e. a new biomarker for patients with Crohn's disease, I did have difficulty in understanding the design and the rationale of the study while I was reading through it. Precisely

We effusively appreciate the constructive criticism of all reviewers that has allowed us to improve the article and we have taken it into account in its entirety. Below we have reviewed the comments made point by point and the subsequent response appears in the order shown

1) By reading the introduction the authors do not mention anything about the experimental part of the animal study. On the contrary they mention "in this study we evaluated the capacity of AIF-1 and CRP to predict the severity of CD. This objective was pursued by the analysis of serum AIF-1 levels in CD patients treated with anti-TNFa blockers. Likewise, the levels of AIF-1 were studied in colon biopsies of patients with CD".

I do not see anything here about the animal models.

We have changed the phrase "Furthermore, we recently demonstrated an increase in tissue levels of AIF-1 in experimental colitis, supporting its role in intestinal inflammation (16)" (Lines 77-78) that appears in the text to:

Action.-Furthermore, we recently demonstrated an increase in tissue levels of AIF-1 in rats with colitis induced with DSS (dextran sodium sulfate) which supports its role in intestinal inflammation (16)".

We hope this will better understand that we have included AIF-1 data in animal models of colitis.

I do not see how the aim of the study was structured and how the authors were planning to prove the relationship of CD with this novel biomarker.

Comment. - DSS-induced colitis in rats is a good model to study the molecular mechanisms involved in human inflammatory bowel disease (Crohn's disease and ulcerative colitis), so we had thought that AIF-1 levels in CD could vary as in the animal model, as it has been.

Action.-  Following the R3 suggestion, we have included the following paragraph in the Introduction:

Since DSS-induced colitis is a good model to study CD in humans, we decided to evaluate the capacity of AIF-1 alone or in combination with CRP (a classical biomarker) to predict the severity of CD.”

2) In the Methods section please provide information about the CD patients type of therapy and in whom were biopsies taken (how many patients since there were not taken in everyone) and with criteria were those patients selected.

…CD patients type of therapy

Comment.- Patients received anti-TNF therapy (infliximab or adalimumab) as described in 2.1 Patients.

All patients were naïve to anti-TNF therapy, were assigned to receive adalimumab (ADA) or infliximab (IFX) according to standard clinical practice, and were followed up for 1 year. Patients received 160/80 mg adalimumab at weeks 0 and 2 and 40 mg at week 6 and every 8 weeks thereafter (14, 22, 30 38, 46, 54 weeks) or infliximab 5 mg/kg at weeks 0, 2, 6 and every 8 weeks thereafter (14, 22, 30, 38, 46, 54 weeks). Patient sera were analyzed at baseline (week 0) and at 14 weeks after the beginning of anti-TNF administration.”

Action.- This comment

…in whom were biopsies taken (how many patients since there were not taken in everyone) and with criteria were those patients selected.

Comment.- In point 3.3 of the results, the number of patients shown and the number of biopsies that were performed are explained. The selection of the patients was random, those with the greatest inflammation were not premeditatedly chosen.

“Figure 4A presents the results from 2 Crohn's patients, which are representative of five who were studied.”

Action.- This comment.

3) I find the Discussion confusing since data from results are presented here. The Discussion has to focus on what data are available about this subject so far and what new this study is bringing to the literature.

In order to give AIF-1 a greater impact as a biomarker, we have included in the discussion the comparative data of the Odds Ratio of CRP, AIF-1 alone and CRP plus AIF-1 in the hope that it can improve the discussion.

Finally, we want to say that we greatly appreciate the very constructive feedback we have received from reviewers.

Round 2

Reviewer 3 Report

No more comments